

# Rapid screening mutations of first-line-drug-resistant genes in *Mycobacterium tuberculosis* strains by allele-specific real-time quantitative PCR

Pengpeng Yang, Yuzhu Song, Xueshan Xia and A-Mei Zhang

Faculty of Life Science and Technology, Kunming University of Science and Technology, Kunming, China

## ABSTRACT

Tuberculosis (TB) is a worldwide health, economic, and social burden, especially in developing countries. Drug-resistant TB is the most serious type of this burden. Thus, it is necessary to screen drug-resistant mutations by using a simple and rapid detection method. A total of 32 pairs of allele-specific PCR (AS-PCR) primers were designed to screen mutation and/or wild-type alleles of 16 variations in four first-line drug-resistant genes (*katG*, *rpoB*, *rpsL*, and *embB*) of TB strains. A pair of primers was designed to amplify 16S rRNA gene and to verify successful amplification. Subsequently, we tested the specificity and sensitivity of these AS-PCR primers. The optimized condition of these AS-PCR primers was first confirmed. All mutations could be screened in general AS-PCR, but only 13 of 16 variations were intuitively investigated by using real-time quantitative PCR (qPCR) and AS-PCR primers. The results of specificity assay suggested that the AS-PCR primers with mutation and/or wildtype alleles could successfully amplify the corresponding allele under optimized PCR conditions. The sensitivity of nine pairs of primers was 500 copy numbers, and the other seven pairs of primers could successfully amplify correct fragments with a template comprising $10^3$ or $10^4$ copy numbers template. An optimized AS-qPCR was established to screen drug-resistant mutations in TB strains with high specificity and sensitivity.

## INTRODUCTION

Tuberculosis (TB) is a disease with high prevalence and mortality rate. It is caused by *Mycobacterium tuberculosis* (*M. tuberculosis*) infection. One-third of the world population is infected with *M. tuberculosis*, and 5% of infected people developed into TB in their lifetime (*Comstock, Livesay & Woolpert, 1974*; *Koul et al., 2011*). In 2017, 10.1 million people suffered from TB, among which 1.6 million people died. Though morbidity and mortality of TB gradually decreased with the appearance of anti-TB drugs, the mutation rate of drug-resistant genes in *M. tuberculosis* seemed to increase recently. The cost of

Corresponding author
A-Mei Zhang, zam1980@yeah.net

TB treatment and research reached $ 10.4 billion in 2018. Half of this amount was used to treat drug-resistant TB (DR-TB) patients (*WHO, 2018*).

In the middle of 20th century, DR-TB strains were reported for the first time (*Crofton & Mitchison, 1948*). Unfortunately, the researchers did not focus on this phenomena at that time (*Zignol et al., 2016*). Recently, the number of patients with DR-TB, especially those with multiple drug-resistant TB (MDR-TB), seriously increased. The numbers of patients with MDR-TB reached 160,684 in 2017. A total of 10,800 cases of extensive drug-resistant TB (XDR-TB) were reported by 77 countries, and 88% XDR-TB cases were from European and South-East Asia regions (*WHO, 2018*). Due to abuse of antibiotics and environmental disruption, DR-TB strains have turned out to be barriers to TB treatment.

First-line anti-TB drugs, including isoniazid, rifampicin (RIF), streptomycin (SM), ethambutol (EMB), and pyrazinamide (PZA), are still widely used in clinics. Inevitably, drug-resistant genes exist in *M. tuberculosis*, thereby allowing it to resist first-line anti-TB drugs. *katG* and *inhA* genes are two common candidate genes in isoniazid (INH) -resistant TB strains. Mutations in other genes, such as *sigI*, *ndh*, and others, also reportedly lead to INH-resistance in TB (*Guo et al., 2006*). Similarly, *rpoB*, *embB*, *rpsL*, and *pncA* were major drug-resistant genes for RIF-, EMB-, SM-, and PZA- resistant TB strains (*Lee et al., 2012*; *Sandy et al., 2002*; *Zhang & Yew, 2009*). Although the mutant spectra showed distinction in different countries, there were some hotspot mutations in these candidate genes. The mutation at codon 315 of the *katG* gene was the most popular INH-resistant mutation, and more than 50% of INH-resistant mutations located in this codon (*Afanas'ev et al., 2007*; *Yuan et al., 2012*). Most of RIF-resistant mutations located in hotspot region of the *rpoB* gene, but the mutant frequency varied from 75% to 90% in different countries (*Franco-Sotomayor et al., 2018*; *Thirumurugan et al., 2015*). Mutations at codon 43 and 88 were two common SM-resistant mutations in the *rpsL* gene, and over 65% SM-resistant TB strains were caused by these two mutations (*Tudo et al., 2010*; *Zhao et al., 2015*). About 70% of EMB-resistant TB strains had mutations located at codon 306, 406, or 497 in the *embB* gene (*Brossier et al., 2015*). Thus, rapid screening for these mutations was necessary.

Long-time and inappropriate drug usage could lead to drug-resistance. Thus, rapid and convenient diagnosis of DR-TB patients is necessary for further effective treatment. Based on the drug-resistant mutations (including hotspot and rare mutations) in our previous study (*Li et al., 2017*), we established an optimized allele-specific real-time quantitative PCR (AS-qPCR) method to screen mutations in DR-TB strains rapidly and with high sensitivity and specificity.

## MATERIALS AND METHODS

### *Mycobacterium tuberculosis* strains collection and DNA extraction

Drug-resistant *M. tuberculosis* strains were collected and cultured in Lowenstein-Jensen medium by doctors in Kunming Third People's Hospital. Drug susceptibility testing (DST) was performed by using the following drugs: INH, 0.2 mg/L; RIF, 40 mg/L; SM, 4.0 mg/L; EMB, 2.0 mg/L; and PZA, 100 mg/L. G$^+$ Bacteria Genomic DNA Kit (ZOMANBIO, Beijing, China) was used to extract genomic DNA from *M. tuberculosis* strains according

to the manufacturer's instructions. This study was approved by the institutional review board of Kunming University of Science and Technology (Approval No. 2014SK027).

## Primer design and AS-PCR optimization

A total of 32 pairs of primers for AS-PCR were designed to screen the 16 variations (including mutation and wild-type alleles), which were located in four genes (*katG*, *embB*, *rpsL*, and *pncA*), by using Oligo Primer Analysis Software v.7 (Table S1). To strengthen the specificity of primers, a non-complementary nucleotide in 3′ end of the allele-specific primer was factitiously changed and marked in red in Table S1. One pair of inner control primers (16S 915-F/16S 1018-R) was used to control and identify PCR quantification (Table S1).

PCR was performed in 20 μL reaction volume and involved 30 ng of genomic DNA, 10 μL 2× TSINGKE™ Master Mix (including one U DNA polymerase, 1.5 mM MgCl$_2$, 50 mM KCl, and 100 mM dNTP) (TSINGKE, Beijing, China) or ChamQ™ SYBR qPCR Master Mix (Vazyme, Nanjing, China), and 0.5 μM each primer (including AS-PCR primers and internal control primers). After optimization, we used the following PCR condition: one cycle of 95 °C for 3 min; 35 cycles of 95 °C for 30 s, optimized temperature for 30 s (Table S1), 72 °C for 10 s; and one extension cycle of 72 °C for 5 min.

## Plasmids construction

Five pairs of primers (Table S2) were designed to amplify fragments containing all mutations. PCR amplification products were ligated into the pClone 007 Blunt Simple Vector by using pClone 007 Vector Kit (TSINGKE, Beijing, China). All constructed plasmids were identified to carry the specific mutation or wildtype allele by sequencing. Plasmid extraction small Kit (TIANGEN, Beijing, China) was used to purify the plasmids.

## Specificity and sensitivity tests

We amplified 14 wildtype TB samples (identified by sequencing) and all templets of each mutation to test the specificity of AS-PCR by using both wildtype and mutation primers. If there was only one sample with a certain mutation, we duplicated the AS-PCR by using the same mutation sample. The sensitivity results were determined based on the appearance and intensity of the products on the agarose gel. Moreover, wildtype AS-PCR primers were also used to amplify plasmids with corresponding mutations. After qualifying the plasmids, we diluted plasmids to $10^4$, $10^3$, and $5 \times 10^2$ copy numbers to achieve a sensitive assay.

## AS real-time quantitative PCR

According to the results of optimized AS-PCR, we performed AS-qPCR assays, in order to directly detect the products and determine the mutation or wild type allele. PCR was performed in 20 μL reaction volume and involved 30 ng of genomic DNA, ChamQTM SYBR qPCR Master Mix (Vazyme, Nanjing, China), and 0.3 μM each primer (including mutation and wild type AS-PCR primers of each allele) on Takara Thermal Cycler Dice Real Time System TP800 (TaKaRa, Kusatsu, Japan).
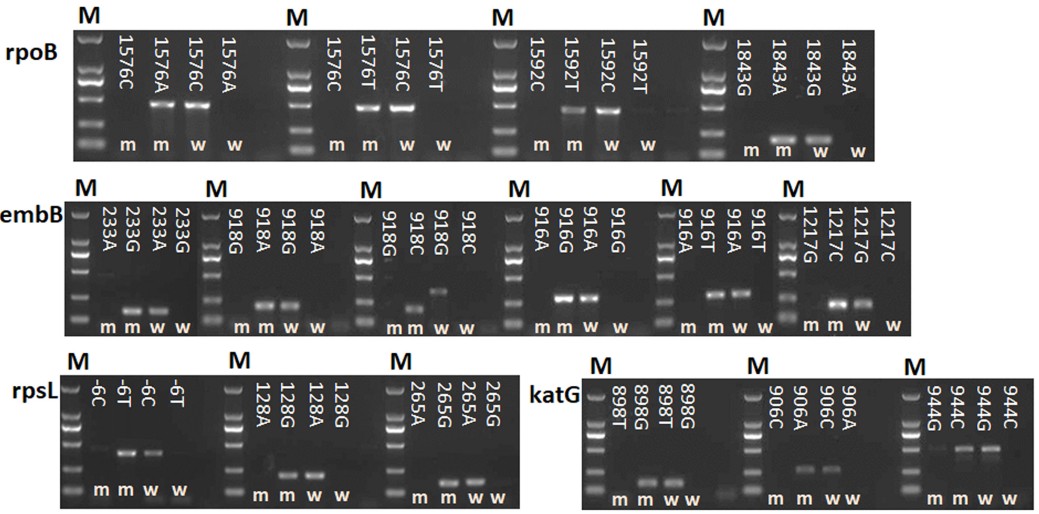

**Figure 1 Detection of 16 mutations in four first-line drug-resistance genes by using AS-PCR.** M means DNA marker DL2000; m and w mean the AS-PCR primers were used to screen mutation and wildtype alleles, respectively. NC means negative control.

# RESULTS

A total of 16 point mutations in four first-line drug-resistant genes were used to establish AS-PCR detecting method, which were identified in 57 drug-resistant *M. tuberculosis* strains by sequencing in our previous study (*Li et al., 2017*). After optimizing the conditions of PCR with these primers, we obtained concordant results by sequencing, that is, the correct bands were successfully amplified by using mutant AS-PCR primers and mutation template but not wild-type samples, and vice versa (Fig. 1). The specificity test results suggested all primers in this study could identify the mutation or wildtype alleles with high fidelity (Fig. S1). As shown in Fig. 2A 206 bp fragment could be amplified by using samples with wild-type *rpsL* gene and primers *RpsL* 128A-F/*RpsL* 128-R (the primer for wild-type allele A at amino acid codon 128). However, this amplicon did not exist when samples with *rpsL* 128G were used. On the contrary, the primers *RpsL* 128G-F/*RpsL* 128-R (for mutant allele G) could amplify a 206 bp fragment when the samples carried the mutation allele 128 G in the *rpsL* gene. The presentation of the 104 bp inner control fragment suggested a successful amplification.

Plasmids with mutation and wildtype alleles were constructed to evaluate the sensitivity of AS-PCR primers. As shown in Fig. 3, the sensitivity of primers for amino acid codon 128 in the *rpsL* gene was estimated by using a plasmid with $10^4$, $10^3$, and $5 \times 10^2$ copy numbers. When the copy number of the plasmids was $5 \times 10^2$, we could visualize a faint band. However, the bands were obvious and clear when we increased the plasmids to $10^3$ or $10^4$ copy numbers. The sensitivity of all 32 pairs of primers were tested, and half of these primers could amplify an observable band when the plasmid copy number was $5 \times 10^2$ (Fig. S2). However, the others needed more copy numbers ($10^3$ or $10^4$, Fig. S2; Table S1).

To directly and rapidly investigate the testing results of AS-PCR, we combined AS-PCR primers and qPCR for subsequent observation. By observing the melt-curve of these products, we determined whether mutations existed. We firstly defined the baseline at 100

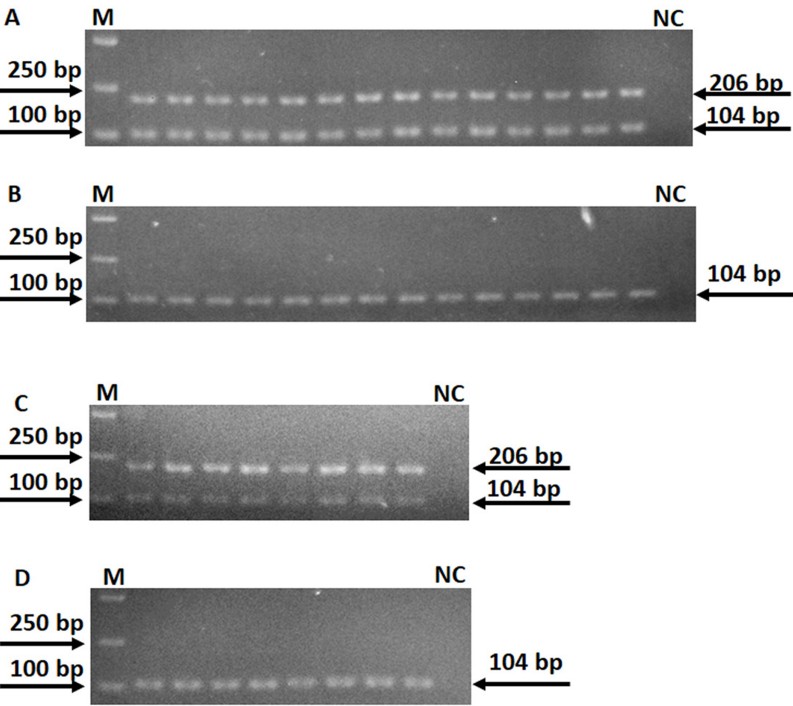

**Figure 2 Specificity of AS-PCR primers for A128G in the *rpsL* gene.** (A) PCR were performed by using 14 DNA templates with allele A and wildtype AS-PCR primers. (B) PCR were performed by using 14 DNA templates with allele A and mutation AS-PCR primers. (C) PCR were performed by using eight DNA templates with allele G and mutation AS-PCR primers. (D) PCR were performed by using eight DNA templates with allele G and wildtype AS-PCR primers. Fragments at 206 bp mean the specific product by AS-PCR primers; fragments at 104 bp mean the inner control product by inner primers; NC means negative control.             

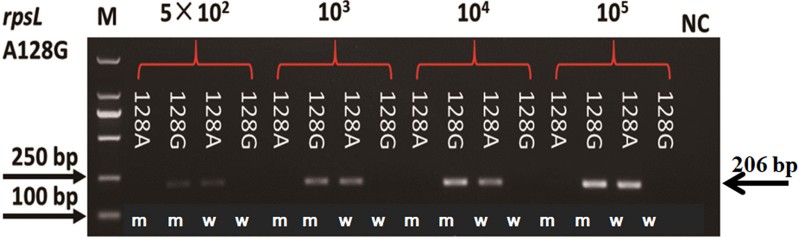

**Figure 3 Sensitivity of AS-PCR primers for A128G in the *rpsL* gene.** M means DNA marker DL2000; m and w mean the AS-PCR primers were used to screen mutation and wildtype alleles, respectively.
             

relative fluorescence units/min as the detecting level. According to this standard, we tested all 16 mutations by using AS-qPCR. However, only 13 mutations were rapidly and directly genotyped (Fig. S3). As shown in Fig. 4, we could rapidly identify the allele at 128 nucleotide in the *rpsL* gene. By using the optimized AS-qPCR, we rapidly distinguished the mutation and/or wild-type allele in one PCR reaction with two pairs of primers.

## DISCUSSION

China is among the top 22 countries with the highest burden of TB and with the second highest burden of DR-TB. About 120,000 persons developed into TB in China each

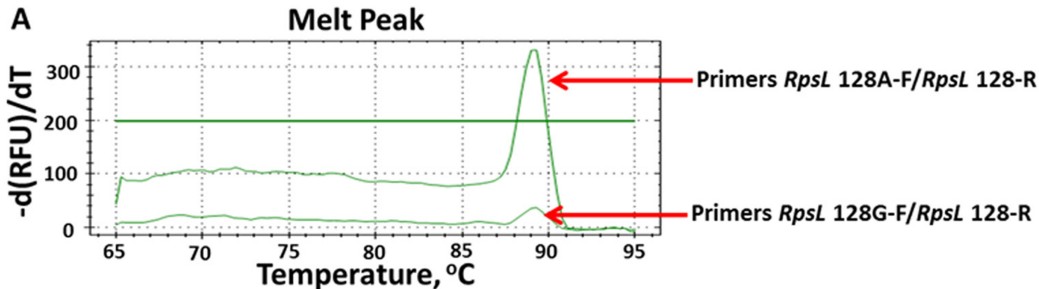

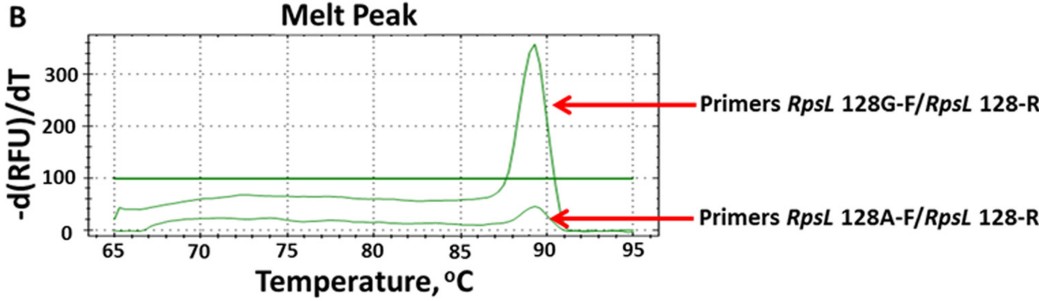

**Figure 4 Detecting allele type of A128G in the *rpsL* genes by using AS-PCR primers and real-time quantitative PCR.** (A) Melt curves of product by using DNA template with allele A. (B) Melt curves of product by using DNA template with allele G.

year (*Du et al., 2017*; *Zhao et al., 2012*). Many factors could lead to DR-TB, such as contaminative environment, drug abuse, long-time therapy, and host genetic factors. Some TB strains might change to drug-resistance or solo DR-TB might develop to XDR-TB after the long-term treatment of TB patients. Thus, it was important and necessary to rapidly diagnose DR-TB, especially for first-line DR-TB. Point mutations of drug-resistant genes were common reasons for the development of first-line DR-TB strains. Mutations in the *katG* and/or *inhA* gene were the two main causes of INH-resistant TB. About 51% and 10% isoniazid-resistant TB strains were caused by mutations in the *katG* and *inhA* genes, respectively (*Guo et al., 2006*). Mutations in the *rpoB* gene, *rpsL* gene, *embB gene*, and *pncA* gene were the main factors for RIF-resistant, SM-resistant, EMB-resistant, and PZA-resistant TB strains, respectively (*Brossier et al., 2015*; *Scorpio & Zhang, 1996*; *Stoffels et al., 2012*; *Villellas et al., 2013*). Furthermore, some hotspot mutations still exist in these drug-resistant genes (*Banerjee et al., 1994*; *Dalla Costa et al., 2009*; *Lee et al., 2012*; *Seifert et al., 2015*; *Waagmeester, Thompson & Reyrat, 2005*). Most commercialized detecting kits for DR-TB strains only contained hotspot mutations. Hence, some rare mutations might be missed. In our previous study, we identified some mutations in these genes of Yunnan DR-TB strains, and some of them were hotspot mutations and others were rare mutations (*Li et al., 2017*). Because no hotspot mutation has been found in the *pncA* gene, 18 mutations of the other four genes (including *katG*, *rpoB*, *rpsL*, and *embB* genes) were used as candidate mutations in this study. Unfortunately, two mutations, including G1388T (at codon 463) in the *katG* gene and A1490G (at codon 497) in the *embB* gene, could not be genotyped

by using AS-PCR. Thus, it seemed that not all drug-resistant mutations could be well detected by using AS-PCR.

Drug susceptibility testing is the classic method and the "gold standard" for the evaluation of DR-TB strains (*Ahmad & Mokaddas, 2009*). Until now, DST is still widely used in laboratory and hospital, but methods to screen mutations of drug-resistant genes were speedily developed after the invention of PCR. Multi-fluorescence qPCR is one of the most common methods to detect RIF- and INH-resistant *M. tuberculosis* (*Peng et al., 2016*). Other technologies, such as whole-genome sequencing (WGS) (*Pankhurst et al., 2016*), high-resolution melt, PCR-single strand conformation polymorphism, and oligonucleotide microarrays (*Caoili et al., 2006*; *Herrmann et al., 2006*; *Pietzka et al., 2009*; *Traore et al., 2006*), have also been widely used to screen the mutations of candidate genes in DR-TB strains. However, these technologies have their advantages and shortcomings. For example, high quality and various mutation types could be detected using WGS, but the expensive equipment and reagent limited its usage.

Since AS-PCR was first reported in 1989 (*Newton et al., 1989*), it has been widely used to screen single nucleotide polymorphisms and mutations. Although AS-PCR is considered as low specificity and sensitivity (*Sharma et al., 2016*), *Onseedaeng & Ratthawongjirakul (2016)* identified mutations of the *gyrA* and *parC* gene in *Escherichia coli* with high sensitivity and specificity by using AS-PCR. Due to its simple operation, low cost, and relatively high specificity and sensitivity, we successfully used AS-PCR to screen 16 mutations in four first-line DR-genes. After optimizing reaction conditions, we successfully amplified the corresponding bands by using AS-PCR primers with high specificity. Most of these primers detected the corresponding mutations when the DNA template reached 500 copy numbers. All these results suggested that optimized AS-qPCR could be used to screen mutations of drug-resistant genes in TB strains with higher specificity and sensitivity. A limitation of this current study was the small size of the number of drug-resistant mutations. One reason was that no more TB-strains with other drug-resistant mutations were obtained in this experiment; another reason was several drug-resistant mutations could not be well genotyped by using the AS-PCR or AS-qPCR method. In further studies, we should collect more DR-TB strains from various regions and further optimize AS-PCR condition for rapid screening.

## CONCLUSION

In summary, we established an optimized AS-qPCR method to screen mutations in four first-line drug-resistant genes of *M. tuberculosis* with relatively high specificity and sensitivity. This AS-qPCR could be widely used in the future to rapidly screen mutations in DR-TB strains.

## ACKNOWLEDGEMENTS

We thank Mr. Daoqun Li for helping extract DNA from TB strains.

### Funding

This study was supported by the National Natural Science Foundation of China (31460289) and Yunnan Science and Technology Commission (2015BC001). The funders had no role in study design, data collection and analysis, decision to publish, or preparation of the manuscript.

### Grant Disclosures

The following grant information was disclosed by the authors:
National Natural Science Foundation of China: 31460289.
Yunnan Science and Technology Commission: 2015BC001.

### Competing Interests

The authors declare that they have no competing interests.

### Author Contributions

- Pengpeng Yang performed the experiments, prepared figures and/or tables, approved the final draft.
- Yuzhu Song analyzed the data, authored or reviewed drafts of the paper, approved the final draft.
- Xueshan Xia contributed reagents/materials/analysis tools, approved the final draft.
- A-Mei Zhang conceived and designed the experiments, analyzed the data, contributed reagents/materials/analysis tools, prepared figures and/or tables, authored or reviewed drafts of the paper, approved the final draft.

### Ethics

The following information was supplied relating to ethical approvals (i.e., approving body and any reference numbers):

The institutional review board of Kunming University of Science and Technology approved this study (Approval No. 2014SK027).

### Data Availability

Raw data is available in the Supplemental Files.

### Supplemental Information

Supplemental information for this article can be found online at http://dx.doi.org/10.7717/peerj.6696#supplemental-information.

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
