# Peer review of "Rapid screening mutations of first-line-drug-resistant genes in Mycobacterium tuberculosis strains by allele-specific real-time quantitative PCR"

_PeerJ, doi:10.7717/peerj.6696_

## Round 0.1 · original submission · Major Revisions

As you can see, your paper was well received but several remarks were raised that overall call for a major revision. If you are willing to make these corrections, we will gladly reexamine your paper. Please, do not for get to add a detailed rebuttal where you explain how (and where in the submission) you have taken these suggestions into consideration and how the paper has been modified.

Reviewer 1 ·

Basic reporting

The manuscript is well written. Some minors errors are present as follows:

1. Line 18: “detection” instead of “detecting”
2. Line 19: AS-PCR is defined on Line 22, yet mentioned here first. Maybe a good idea to define it here.
3. Line 25: “Specificity” not “Specific”
4. Line 27-29: The meaning of the sentence is not conveyed clearly. Maybe consider re-writing it.
5. Line 41: “$” instead of “dollars”
6. Line 70: use of upper case letters in the name of the hospital “Kunming Third People’s Hospital” and not “Kunming third People’s Hospital”
7. Line 74: “a total” not “totally”
8. Lines 74-76: These seem to be out of place here. Maybe they need to be moved into the end of the introduction or to the beginning of the results to validate why the authors chose to create primers for the specific alleles they studied.
9. Line 85: “in” not “at”
10. Line 86: In the description of the contents of 2×TSINGKE master mix, the amounts and the units are after the component “….(….DNA polymerase 1 U,MgCl2 1.5 mM….)…”. Conventionally, the amounts are written before the units, for example”….(…. 1 U DNA polymerase,1.5 mM MgCl2…..)…”
11. Line 108: “…..optimizing the conditions of these primers….” should probably read “….optimizing the conditions of PCR with these primers….”
12. Line 112: does “high quality” mean “high fidelity”? If not, the authors should define what it means.
13. Also check spacing between words and brackets.

Use of references:

1. Line 107: A reference seems to be missing here.

Comments about figures and tables:

1. Figures 4 and S2: The numbers on the axes and the labels are too small.
2. Figure 2: Why does each gel have so many samples? There is no explanation in the text or the figure legend.
3. Figure S1: the lower bracket with the size of the product is confusing. Could the authors use arrows instead as they have done for the markers?
4. Tables: check spacing between words

Experimental design

The experiments are well described. However, there are some aspects that could use more information. My comments are as follows for the Methods section of the manuscript:

1. Lines 103-104: A sensitivity assay is mentioned, but there is no description of how the sensitivity was assessed. The text in the results suggests that the sensitivity was determined based on the appearance and intensity of the product on the gel. One sentence stating this would be helpful here.

2. A real-time PCR was performed. However, there is no description of the method. A method description mentioning the templates, the primers, their respective concentrations and the reagents used would be helpful in reproducing the results.

Validity of the findings

The results are presented well and the conclusions are well described. However, in Line 27, the authors claim "....sensitivity of third-quarters primers was 500 copy numbers....", whereas in Line 124 and 125 it is said "....half of these primers could amplify an observable..." . Could the authors clarify which one is correct?

Additional comments

No comment

·

Basic reporting

The present work "Rapidly screening mutations of first-line-drug-resistant genes in Mycobacterium tuberculosis strains by allele-specific real-time quantitative PCR" is a methodologically well conducted study addressing an important topic on the field of tuberculosis research as the design of faster and accurate methods for drug resistance profiling of strains is. The authors described a new protocol based or single nucleotide polimorphysm detection by real time PCR from M. tuberculosis isolates.
However, I have several major concerns about how this work is presented on the current version:
1. The information provided as background on the introduction is really poor. On lines 55-60 the authors simple describe the genes responsible for drug resistance but do not detailed the main mutations described worldwide that explain drug resistance on TB isolates. For instance, KatG 315 mutation is responsible for approximately 60% of isoniazide resistance and mutations 531, 516 and 526 at hotspot on rpoB are responsible for 90% or more of rifampicine resistance. This information is crucial to de included for all he 4 genes on this study.
2. It is only clear to me which mutations and why were included on this study after reading a reference (Li et al, 2017) detailed only on methods. Results section (lines 107 and 108) starts with authors saying that 16 mutations included on this work were described on a previous study, that must me cited again. On that paper, a quite unique mutation frequency distribution is described by the same authors on this paper for the Yunnan region of China. I am assuming that based on that findings those mutations where the one selected to develop this new protocol. However, this point is mandatory to be address on this manuscript introduction and discussion sections, otherwise the reader will not understand why these mutations were selected when they are not the most common one worldwide. So far, a table with the codon location for all the mutations included should be somewhere on the manuscript (I am assuming the mutations codon are the one on the name of the primers on supplementary figure 1). Additionally, the authors need to explain what proportion of each kind of drug resistance is caused by the mutations included on the study. Most common mutations like the detailed on paragraph above were not selected so the future impact of this new protocol would be limited to the Yunnan region and that should be said. Some kind of comparison and contextualization with Gene-Xpert is missing too, at least regarding mutations on rpoB, and how improve "local" protocols based on particularly epidemiological scenarios like on Yunnan are neccessary.
3. The articule structure needs to be deeply changed. While on figure 1 all the genes studied are included it looks like the authors only show one of the mutation on rpsL as "an example" or something like that, while the rest of the mutations on rpsL and even the rest of the genes are included as supplementary material. This makes the article really difficult to read. Additionally, fundamental data simply cannot be presented as supplementary material. Instead of showing 14 samples run for rspL mutation, authors can show in figures on the paper 1 or 2 samples for each mutation of each gene and then a supplementary figure for the 14 samples for each mutation and each gene.
4. No clear explanation is made of why only 13 of the 16 mutations are tested on the real time PCR protocols. On line 24 of the abstract, authors says that "intutuively" they selected 13 mutations. What do the mean by "intutitevely'?

Additionally, other minor comments:
1. I will recommend to review grammar and vocabulary on the whole text by a native speaker. I am not a native english speaker but I believe there are corrections that needs to me made. For example, I believe the title is wrongly written, and instead of "rapidly screening" it should be "rapid screening". On line 48, "persons" is incorrect, it should be people.
2. In the abstract, line 27, instead of saying "third quarters" and "the others", please simple say the number of mutations with the different copy number level of mutations. Otherwise sounds not accurate.
3. On the introduction, line 41. It is not possible that 8 million dollars is the worldwide expense on TB. 8 billion perhaps? Please correct.
4. Please, update epidemiology TB data used on introduction using World TB report from 2018 or at least 2017. But authors are citing 2015 report.
5. Figure design might be improved. Use of red markers is unnecesary. Instead of 1 and 2 directly use wt (wild type) and mut (mutants). No need of M for DNA ladders and no need to include DL2000 ladder on the legend.
6. Figure 1. NC is not stated on the figure legend as Negative control.
7. Supplementary material. Figure and table legends are poorly detailed.

Experimental design

The experimental design is methodologically well address despite the lack of conceptualization described on the "Basic reporting" section. But several minor comments deserve to be done:
1. Better description of samples used on the study. Pulmonary and extrapulmonary? Age range? Also I believed that the 14 samples used must be the ones described and not the 57 isolates used for 16 mutations identification on the previous paper (Li et al, 2017).
2. Line 82. When describing primer designing, at 3', "non complementary nucleotide" is the correct term better than "random".
3. On figure 2, why 14 samples are included on panels A and B but only 7 in panels C and D?
4. On supplementary figure, for KatG mutations not always the copy number tested were the same for the different mutations. Is there any reason for that? For the other genes, all the mutations are tested with same copy numbers alternatives.

Validity of the findings

Regarding the validity of the findings, although data is robust and conclusion well stated, like in the introduction, results are not properly discussed and contextualized. All the comments made on point 1 and 2 at the "Basic reporting" section should be taking in consideration for rewriting the discussion.

---

## Round 0.2 · Minor Revisions

The corrections requested by one reviewer concern the spelling. Please, have your paper thoroughly checked for spelling mistakes, preferably by a native English speaker. This is important as PeerJ does not provide copy-editing service and all spelling mistakes in your submission will be made public... which is not what you may like to see.

Reviewer 1 ·

Basic reporting

The authors have addressed all the concerns raised previously. Please check all spellings.

Experimental design

no comment

Validity of the findings

no comment

·

Basic reporting

The authors have properly address on the response letter and on the manuscript all the major and minor concerns detailed on the first review. Introduction and discussion have been extraordinarly improved.

Experimental design

The authors have properly address on the response letter and on the manuscript all the major and minor concerns detailed on the first review regarding the experimental design.

Validity of the findings

Discussion has being greatly improved following reviewers suggestions allowing readers a better conceptualization of the importance of this research findings.

---

## Round 0.3 · accepted · Accept

Thank you for making the last corrections. I believe the paper is now acceptable for publication.

#